# Simple and Expedient Access to Novel Fluorinated Thiazolo- and Oxazolo[3,2-*a*]pyrimidin-7-one Derivatives and Their Functionalization via Palladium-Catalyzed Reactions

**DOI:** 10.3390/molecules27093013

**Published:** 2022-05-07

**Authors:** Wafa Blancou, Badr Jismy, Soufiane Touil, Hassan Allouchi, Mohamed Abarbri

**Affiliations:** 1Laboratoire de Physico-Chimie des Matériaux et des Electrolytes pour l’Energie (PCM2E), EA 6299, Avenue Monge, Faculté des Sciences, Université de Tours, Parc de Grandmont, 37200 Tours, France; blancowafa@gmail.com; 2Laboratoire des Composés Hétéro-Organiques et Matériaux Nanostructurés (LR18ES11), Faculté des Sciences de Bizerte, Université de Carthage, Zarzouna 7021, Tunisia; soufiane.touil@fsb.rnu.tn; 3Faculté de Pharmacie, Université de Tours, EA 7502 SIMBA, 31 Avenue Monge, 37200 Tours, France; hassan.allouchi@univ-tours.fr

**Keywords:** thiazolo[3,2-*a*]pyrimidin-7-ones, oxazolo[3,2-*a*]pyrimidin-7-ones, fluorinated alkynes, regioselective, one-pot reaction, Suzuki-Miyaura, Sonogashira coupling

## Abstract

An efficient, versatile, and one-pot method for the preparation of novel fluorinated thiazolo- and oxazolo[3,2-*a*]pyrimidin-7-ones is described from 2-aminothiazoles or 2-amino-oxazoles and fluorinated alkynoates. This transformation, performed under transition-metal-free conditions, offers new fluorinated cyclized products with good to excellent yields. Moreover, the functionalization of these *N*-fused scaffolds via the Suzuki-Miyaura and Sonogashira cross-coupling reactions led to the synthesis of highly diverse thiazolo- and oxazolo[3,2-*a*]pyrimidin-7-ones.

## 1. Introduction

Thiazolo[3,2-*a*]pyrimidines are an important class of heterocyclic compounds that have aroused intensive research interest from chemists due to their wide range of biological properties, including antipsychotic [1], anticancer [2,3], anti-inflammatory [4,5,6,7], antimicrobial [8,9], antiviral [10,11] and anti-HIV activities [12]. Additionally, many compounds containing the oxazolo[3,2-*a*]pyrimidine framework are known for their pharmacological potential as anti-inflammatory [13], antihypertensive [14], and anti-leukemia agents [15]. It should be noted that some examples of structural analogues of thiazolo- and oxazolo[3,2-*a*]pyrimidinones, are used in clinical medicine, such as Ritanserin and Setoperone, which are antipsychotics [16], and SAR218645, which is an mGluR2 positive allosteric modulator that has been shown to be effective in treating some aspects of cognitive dysfunction in schizophrenia (Figure 1) [17].

In recent years, fluorinated heterocycles have acquired a crucial role in the pharmaceutical and agrochemical industries due to their wide range of biological properties [18,19]. In addition, fluorinated compounds are present in about 20% of drugs [20,21,22]. In general, the introduction of fluorine atoms or fluoroalkyl (perfluoroalkyl) groups into heterocyclic compounds increases lipophilicity, stability, solubility, reactivity, and biological properties [23,24,25,26,27]. Accordingly, the combination of two pharmacophoric entities such as the thiazolo- or oxazolo[3,2-*a*]pyrimidin-7-one scaffold and perfluoroalkyl groups could be an effective method to enhance the biological activity of these heterocycles.

Classically, Thiazolo and oxazolo[3,2-*a*]pyrimidin-7-one derivatives have been synthesized by reactions of the corresponding 2-aminoazoles with a symmetric alkyne such as dialkyl acetylenedicarboxylates [28,29,30,31,32,33,34,35,36] (Figure 2a), or an asymmetric alkyne such as ethyl propiolate derivatives [37,38,39,40] (Figure 2b). However, despite their relevance, these procedures have several drawbacks such as a lack of generality, poor regioselectivity, the use of expensive reagents or less available reagents, as well as unsatisfactory yields.

Despite the numerous approaches described in the literature for the synthesis of non-fluorinated thiazolo- or oxazolo[3,2-*a*]pyrimidin-7-one derivatives, the incorporation of fluorinated groups into these scaffolds has never been reported to date. Thus, the development of a simple and efficient method using readily available starting materials to access new fluorinated thiazolo- and oxazolo[3,2-*a*]pyrimidin-7-one derivatives is highly desired.

Continuing our interest in the development of efficient methods for the synthesis of new fluorinated heterocyclic compounds with possible biological properties [41,42,43,44,45,46,47], we report here a simple and regioselective synthesis of a novel series of fluorinated thiazolo and oxazolo[3,2-*a*]pyrimidin-7-one derivatives, by [3+3] cyclocondensation of 2-aminothiazoles or 2-amino-oxazoles with fluorinated alkynoates (Figure 2c). This simple synthetic strategy, which relies on the use of easily prepared fluorinated alkynoates [48] and of commercially available 2-aminothiazoles and oxazoles, represents a highly efficient one-step route and regioselective access to fluorinated thiazolo- and oxazolo[3,2-*a*]pyrimidin-7-ones.

## 2. Results and Discussion

In order to find the optimal reaction conditions for the synthesis of the target compounds, 2-aminothiazole, **1a**, and ethyl 4,4,4-trifluorobut-2-ynoate, **2a**, were selected as model substrates for the development of this condensation/heterocyclisation reaction by varying different conditions (solvents, temperature, and catalyst). The reaction conditions investigated leading to thiazolo[3,2-*a*]pyrimidin-7-one **3a** are summarized in Table 1.

First, the reaction was carried out using a variety of solvents without any catalyst. The use of a non-polar solvent, such as toluene, gave only traces of the desired product **3a** (Table 1, entry 1). The use of aprotic polar solvents such as DCE, THF, 1,4-dioxane, MeCN, DMF, and DMSO resulted in very low to moderate yields of **3a** (Table 1, entries 2–7). Switching to a protic polar solvent such as H_2_O did not provide any improvement in the conversion of the starting material, and the expected product **3a** was isolated with a yield not exceeding 28% (Table 1, entry 8), presumably due to the low solubility of the starting materials in water. In contrast, a significant improvement in the yield of target product **3a** occurred, reaching 72%, when the reaction was performed with ethanol as the reaction solvent (Table 1, entry 9). To our delight, when the cyclocondensation reaction was run in MeOH as a solvent, the expected product was obtained with a good yield reaching 88% (Table 1, entry 10). Increasing the reaction temperature to 100 °C led to a decrease in the yield of compound **3a** (Table 1, entry 11). At room temperature and under the same conditions, the reaction proceeded with a very low yield and incomplete conversion of **1a** after 72 h (Table 1, entry 12). Further optimizations were undertaken using various catalysts, including AgSO_3_CF_3_, Ag_2_CO_3_, AgOAc, Cu(OAc)_2_, Pd(Oac)_2_, ZnCl_2_, and CuBr, but did not improve the reaction yield and provided the cyclized product **3a** with yields not exceeding 70% (Table 1, entries 13–19). The molecular structure of the new compound **3a** was unambiguously confirmed by X-ray crystal analysis, as depicted in Figure 3 [49].

With the optimized conditions in hand [**1a** (1 equiv), **2a** (1.3 equiv), MeOH, 70 °C, 12 h], the substrate scope and limitation of the [3+3] cyclocondensation reaction were investigated using various 2-amino-thiazole or oxazole derivatives and fluorinated alkynes. The results are summarized in Figure 1. As shown in Figure 1, activated alkynes substituted with different fluorinated groups (CF_3_, C_2_F_5_, CF_2_Ar) reacted efficiently with 2-aminothiazole **1a** to generate novel fluorinated thiazolo[3,2-*a*]pyrimidin-7-ones **3a-d** with yields ranging from 65 to 88%. It is noteworthy that the presence of an electron-withdrawing atom such as bromine at the 4-position of 2-aminoathiazole did not prevent the success of this condensation/lactamization reaction, providing the brominated cyclized products **3e****-****g** with yields of 63%, 67%, and 62%, respectively. These brominated compounds could serve as key intermediates to access novel functionalized thiazolo[3,2-*a*]pyrimidin-7-ones.

The synthetic scope of this reaction was also successfully extended to benzo[*d*]thiazol-2-amine derivatives to access new fluorinated tricyclic compounds **3h****-****i** in good yields (61 and 60%, respectively). It should be noted that introducing a substituent such as a bromine atom on the aromatic ring of the benzo[*d*]thiazol-2-amine had a negligible effect on the reactions, offering the possibility of diversifying the range of benzo[*d*]thiazolo[3,2-*a*]pyrimidin-7-ones.

To further extend the synthetic scope of this reaction, we sought to examine the reactivity of other 1,3-bis nucleophile reagents, such as 2-amino oxazole derivatives. Interestingly, 2-amino-oxazole derivatives were also efficiently cyclocondensed with fluorinated alkynoates (substituted with a CF_3_, C_2_F_5_, or CF_2_-Ar group) to give the expected products **3j-l** with a yield ranging from 60 to 68%. Continuing the evaluation of the cyclocondensation process, treatment of 2-aminobenzo-oxazole derivatives with ethyl 4,4,4-trifluorobutynoate under the same reaction conditions provided the corresponding oxazolo[3,2-*a*]pyrimidin-7-ones **3m** and **3n** with yields of 61% and 56%, respectively. Again, the presence of a chlorine atom on the aromatic ring had no significant influence on the efficiency of the reaction. It is noteworthy that ^1^H, ^19^F, and ^13^C NMR spectrum analysis of the crude mixture of all examples confirmed that no trace of the second regioisomer is observed, showing full regioselectivity of the [3+3] cyclocondensation process. Thus, this synthetic process appears efficient and versatile for the regioselective synthesis of new thiazolo- and oxazolo[3,2-*a*]pyrimidin-7-ones **3a-n** bearing fluoroalkyl groups.

Functionalization via the Palladium-catalyzed cross-coupling reaction of the C-Br bond at position 2 of compounds **3e****-****f** will allow the preparation of a large library of novel thiazolo[3,2-*a*]pyrimidin-7-ones with a high structural diversity [50,51,52]. First, compounds **3e****-****f** were subjected to Suzuki–Miyaura cross-coupling under known standard reaction conditions [53,54,55] using 1.2 equiv of boronic acid, 10 mol% of PdCl_2_(PPh_3_)_2_, and 2 equiv of Na_2_CO_3_ in a 1,4-dioxane/water (4/1) mixture at 80 °C for 1 h. Under these conditions, these coupling reactions allowed complete conversion of the starting materials and efficient access to the new 2-arylated 5-fluorinated thiazolo[3,2-*a*]pyrimidin-7-ones **4a****-****l** (Figure 2).

As shown in Figure 2, different boronic acids bearing electron-donating or electron-withdrawing groups on the aromatic ring provided the arylated products **4a****-****l** in good to excellent yields. Notably, phenylboronic acid was successfully coupled with compound **3e** leading to the arylation product **4a** with a yield of 91%. Arylboronic acids bearing electron-donating groups, such as a methoxy group in the *ortho*, *meta*, or *para*-position, were readily coupled with **3e** to provide the corresponding products **4b** (82%), **4c** (92%), and **4d** (95%), respectively. It should be noted that a slight drop in yield was observed in the case of compound **4b**, presumably due to the steric hindrance generated by the methoxy group on the *ortho*-position. As expected, the coupling reaction of compound **3f** carrying a pentafluoroethyl group with *p*-methoxyphenylboronic acid led to the arylated product **4e** with a moderate yield of 55%. This result seems to indicate that the nature of the fluoroalkyl group plays a significant role in the efficiency of this coupling reaction. Moreover, the coupling reactions of compound **3e** with 1,4-dimethoxyboronic acid and 2,3-ethylenedioxyboronic acid were easily converted to the desired products **4f** and **4g** with yields of 98% and 60%, respectively. Interestingly, it is worth noting that the reaction was tolerant of the free amino group at the *meta* position of the phenylboronic acid, providing the corresponding compound **4h** in a yield of 52%. Likewise, phenylboronic acid substituted at the *para* or *meta* position with an electron-withdrawing group such as CF_3_ yielded the expected products **4i** (75%) and **4j** (57%), respectively. Gratifyingly, the coupling reaction was extended to a heteroarylboronic acid such as 3-thiophenyl, providing the cross-coupling product **4l** in excellent yield (95%).

To showcase other practical applications of cross-coupling reactions and in order to introduce more molecular diversity at the C-2 position of the 2-bromo thiazolo[3,2-*a*]pyrimidin-7-ones **3e****-****f**, we extended our work to Sonogashira cross-coupling reactions [56,57], which are a powerful method for the formation of new CSp-CSp^2^ bonds (Figure 3). Alkynylated heterocycle compounds are, however, prevalent in many bioactive and natural products with a wide range of interesting biological activities [58,59,60,61]. In the context of synthetic transformations, these compounds allow a broad range of transformations, including click reactions, Michael addition, oxidations, and reductions [62,63,64].

To this end, the brominated compound **3e** and a variety of available terminal alkynes were subjected to standard coupling conditions [alkyne (1.5 equiv), PdCl_2_(PPh_3_)_2_ (5 mol%), CuI (10 mol%), Et_3_N (2 equiv), DMF, 80 °C, 2 h]. The palladium-catalyzed alkynylation reactions of **3e** proceeded successfully, providing the 2-alkynylated-5-trifluoromethyl thiazolo[3,2-*a*]pyrimidin-7-ones **5a****-****h** in good to excellent yields (55–92%) (Figure 3). For example, phenylacetylene was cross-coupled to give the expected product **5a** in a 68% yield.

Additionally, phenylacetylene substituted at the 4-position with an electron-donating group, such as methoxy, generated the coupled product **5b** in a yield of 65%. Similarly, coupling with 4-chlorophenylacetylene provided the desired product **5c** in 58% of yield, showing that an electron-withdrawing group at position 4 of the aromatic ring is also tolerated. We also briefly investigated the effect of the position of the fluorine atom on the aromatic ring on the efficiency of the coupling reaction. Overall, the electron-withdrawing atom fluor at the *para* or *ortho* position was tolerated to give rise to the alkynylated products **5d** (85%) and **5e** (68%), respectively, although a lower yield was observed when the fluorine atom was positioned in the *ortho* position, which is probably due to steric hindrance. Finally, when aliphatic alkynes, such as 1-hexyne and ethynylcyclohexane were subjected to the standard reaction conditions, the expected products **5g** and **5h** were obtained in yields of 75% and 55%, respectively.

As shown in Figure 1, Figure 2 and Figure 3, a wide variety of fluorinated thiazolo- and oxazolo[3,2-*a*]pyrimidin-7-ones were efficiently prepared with a large substrate scope and with good to excellent yields (34 examples). All products are new and have been fully characterized by ^1^H, ^19^F, and ^13^C NMR spectroscopy and HRMS.

Some synthesized derivatives were subjected to in vitro evaluation on human hMAO-A and hMAO-B by using *p*-tyramine as a nonspecific substrate. Screening of the compounds was done at 100 µM and 10 µM, and compounds with residual activities (RAs) below 50% at 100 µM were subjected to IC_50_ determination using a serial dilution of the inhibitors. The results expressed in IC_50_ showed that compound **3g** selectively inhibited human monoamine oxidase A (hMAO-A) with an IC_50_ value of 54.08 μM (Table 2).

## 3. Materials and Methods

### 3.1. General Information

All reactions were performed under an inert atmosphere of argon in oven-dried glassware equipped with a magnetic stir bar. Solvents for reactions were obtained from Thermo Fisher Scientific in extra dry quality and stored under argon over activated 3 Å sieves. All reagents were purchased from Fluorochem and used as received without additional purification. Reactions were monitored by thin-layer chromatography (TLC) analysis using silica gel 60 F254 plates. All products were visualized by exposure to UV light (longwave at 365 nm or shortwave at 254 nm). Column chromatography was performed using silica gel 60 (230–400.13 mesh, 0.040–0.063 mm). Eluents were distilled by the standard methods before each use. All new compounds were characterized by NMR spectroscopy (^1^H, ^19^F, and ^13^C), high-resolution mass spectroscopy (HRMS), and melting point (if solids). NMR spectra were recorded at 300 MHz for ^1^H, 282 MHz for ^19^F, and 75 MHz for ^13^C with a Bruker^®^ 300 MHz NMR spectrometer. Proton and carbon magnetic resonance spectra (^1^H NMR and ^13^C NMR) were recorded using tetramethylsilane (TMS) as an external standard and CDCl_3_ (7.28 ppm for ^1^H NMR and 77.04 ppm for ^13^C NMR) or DMSO-*d6* (2.50 ppm for ^1^H NMR and 40.0 ppm for ^13^C NMR) as internal standards. ^19^F spectra were unreferenced. Data for NMR are reported as follows: chemical shift (δ ppm), multiplicity (s = singlet, d = doublet, t = triplet, q = quartet, sep = septet, m = multiplet and br = broad resonance) and coupling constants *J* are reported in Hertz (Hz). All the NMR spectra were processed in MestReNova. HRMS experiments were performed on a hybrid tandem quadrupole/time-of-flight (Q-TOF) instrument, equipped with a pneumatically assisted electrospray (Z-spray) ion source (Micromass, Manchester, UK) operated in the positive mode. 

### 3.2. General Procedure for the Synthesis of Thiazolo- and Oxazolo[3,2-a]pyrimidin-7-ones 3

To a flame-dried flask containing a magnetic stir bar under an inert atmosphere of argon was added to 2-aminothiazole or 2-amino-oxazole derivatives at 1 mmol (1.0 equiv) and dry methanol (4 mL). The solution was cooled to 0 °C followed by dropwise addition of fluorinated alkyne 1.3 mmol (1.3 equiv). The mixture was allowed to warm to room temperature and then heated at 70 °C for 12 h. The progress of the reaction was monitored by TLC (PE/EtOAc, 70/30). After cooling to room temperature, the solvent was evaporated in vacuo. The crude product was then triturated, rinsed, and washed with Et_2_O. The resulting precipitate was collected via filtration to give the corresponding fluorinated substrates **3a-n** as analytically pure solid without further purification. All compounds **3a-n** were synthesized by adopting this procedure. (^1^H-NMR and ^13^C-NMR of compounds **3a-n** are shown in Appendix A.)

5-(Trifluoromethyl)-7*H*-thiazolo[3,2-*a*]pyrimidin-7-one (**3a**).

Compound **3a** was obtained as a white solid with a yield of 88%. m.p. 210–212 °C; ^1^H NMR (300 MHz, CDCl_3_): δ 7.42 (d, *J* = 5.1, 1H), 7.14 (d, *J* = 5.1 Hz, 1H), 6.76 (s, 1H); ^19^F NMR (282 MHz, CDCl_3_): δ −67.95; ^13^C NMR (75 MHz, CDCl_3_): δ 166.6, 165.4, 135.3 (q, *J* = 37.4 Hz), 121.5 (q, *J* = 3.4 Hz), 119.1 (q, *J* = 274.5 Hz), 111.7 (q, *J* = 4.0 Hz), 111.6; HRMS (ESI) *m*/*z* [M+H]^+^ calcd for C_7_H_4_F_3_N_2_OS: 220.9991; found: 220.9988.

5-(Pentafluoroethyl)-7*H*-thiazolo[3,2-*a*]pyrimidin-7-one (**3b**).

Compound **3b** was obtained as a yellow solid with a yield of 85%. m.p. 181–183 °C; ^1^H NMR (300 MHz, CDCl_3_): δ 7.48 (d, *J* = 5.1 Hz, 1H), 7.00 (d, *J* = 5.1 Hz, 1H), 6.81 (s, 1H); ^19^F NMR (282 MHz, CDCl_3_): δ −82.43, −115.53; ^13^C NMR (75 MHz, CDCl_3_): δ 167.1, 164.9, 134.7 (t, *J* = 27.1 Hz), 122.3 (td, *J* = 7.1, 1.7 Hz), 118.0 (qt, *J* = 287.4, 36.1 Hz), 114.7 (t, *J* = 5.7 Hz), 111.0, 110.0 (tq, *J* = 258.8, 40.9 Hz); HRMS (ESI) *m*/*z* [M+H]^+^ calcd for C_8_H_4_F_5_N_2_OS: 270.9959; found: 270.9954.

5-[(4-Methoxyphenyl)difluoromethyl]-7*H*-thiazolo[3,2-*a*]pyrimidin-7-one (**3c**).

Compound **3c** was obtained as a yellow solid with a yield of 65% d. m.p. 148–150 °C; ^1^H NMR (300 MHz, CDCl_3_): δ 7.54 (dt, *J* = 5.1, 1.7 Hz, 1H), 7.48 (d, *J* = 8.8 Hz, 2H), 7.03 (d, *J* = 8.8 Hz, 2H), 7.87 (d, *J* = 5.1 Hz, 1H), 6.30 (s, 1H), 3.89 (s, 3H); ^19^F NMR (282 MHz, CDCl_3_): δ −93.40; ^13^C NMR (75 MHz, CDCl_3_): δ 166.8, 166.6, 162.3, 142.1 (t, *J* = 32.2 Hz), 127.7 (t, *J* = 5.6 Hz), 123.2 (t, *J* = 25.9 Hz), 123.1 (t, *J* = 5.3 Hz), 118.0 (t, *J* = 243.2 Hz), 114.6 (2C), 114.5 (d, *J* = 106.9 Hz), 113.0 (t, *J* = 4.9 Hz), 109.8, 55.6; HRMS (ESI) *m*/*z* [M+H]^+^ calcd for C_14_H_11_F_2_N_2_O_2_S: 309.0504; found: 309.0496.

5-[(4-Bromophenyl)difluoromethyl]-7*H*-thiazolo[3,2-*a*]pyrimidin-7-one (**3d**).

Compound **3d** was obtained as a yellow solid with a yield of 74%. m.p. 202–204 °C; ^1^H NMR (300 MHz, CDCl_3_): δ 7.72 (d, *J* = 8.5 Hz, 2H), 7.55 (dt, *J* = 5.1, 1.8 Hz, 1H), 7.46 (d, *J* = 8.5 Hz, 2H), 6.92 (d, *J* = 5.1 Hz, 1H), 6.22 (s, 1H); ^19^F NMR (282 MHz, CDCl_3_): δ −94.91; ^13^C NMR (75 MHz, CDCl_3_): δ 166.9, 166.2, 141.3 (t, *J* = 31.1 Hz), 132.7 (3C), 130.3 (t, *J* = 25.9 Hz), 127.7 (t, *J* = 5.7 Hz), 126.9 (t, *J* = 2.1 Hz), 122.9 (t, *J* = 5.3 Hz), 117.7 (t, *J* = 244.1 Hz), 113.4 (t, *J* = 4.9 Hz), 110.1; HRMS (ESI) *m*/*z* [M+H]^+^ calcd for C_13_H_8_F_2_BrN_2_OS: 356.9503; found: 356.9488.

2-Bromo-5-(trifluoromethyl)-7*H*-thiazolo[3,2-*a*]pyrimidin-7-one (**3e**).

Compound **3e** was obtained as a white solid with a yield of 63%. m.p. 184–186 °C; ^1^H NMR (300 MHz, DMSO-*d_6_*): δ 8.19 (s, 1H), 6.81 (s, 1H); ^19^F NMR (282 MHz, DMSO-*d_6_*): δ −67.10; ^13^C NMR (75 MHz, DMSO-*d_6_*): δ = 166.9, 165.0, 135.0 (q, *J* = 37.0 Hz), 124.1 (q, *J* = 2.9 Hz), 119.3 (q, *J* = 274.5 Hz), 111.7 (q, *J* = 4.0 Hz), 101.0; HRMS (ESI) *m*/*z* [M+H]^+^ calcd for C_7_H_3_BrF_3_N_2_OS: 298.9096; found: 298.9093.

2-Bromo-5-(pentafluoroethyl)-7*H*-thiazolo[3,2-*a*]pyrimidin-7-one (**3f**).

Compound **3f** was obtained as a yellow solid with a yield of 67%. m.p. 171–173 °C; ^1^H NMR (300 MHz, CDCl_3_): δ 7.46 (s, 1H), 6.74 (s, 1H); ^19^F NMR (282 MHz, CDCl_3_): δ −82.37, −115.54; ^13^C NMR (75 MHz, CDCl_3_): δ 166.8, 164.1, 134.3 (t, *J* = 27.1 Hz), 122.3 (td, *J* = 7.2, 1.8 Hz), 117.9 (qt, *J* = 251.5, 36.0 Hz), 114.8 (t, *J* = 5.8 Hz), 109.8 (tq, *J* = 259.1, 41.0 Hz), 101.7; HRMS (ESI) *m*/*z* [M+H]^+^ calcd for C_8_H_3_BrF_5_N_2_OS: 348.9064; found: 348.9057.

2-Bromo-5-[(4-bromophenyl)difluoromethyl]-7*H*-thiazolo[3,2-*a*]pyrimidin-7-one (**3g**).

Compound **3g** was obtained as a beige solid with a yield of 62%. m.p. 218–220 °C; ^1^H NMR (300 MHz, CDCl_3_): δ 7.73 (d, *J* = 8.7 Hz, 2H), 7.58 (t, *J* = 1.8 Hz, 1H), 7.45 (d, *J* = 8.7 Hz, 2H), 6.09 (s,1H); ^19^F NMR (282 MHz, CDCl_3_): δ −94.89; ^13^C NMR (75 MHz, CDCl_3_): δ 166.6, 165.5, 141.0 (t, *J* = 31.1 Hz), 132.7 (3C), 129.9 (t, *J* = 25.6 Hz), 127.7 (t, *J* = 5.7 Hz), 127.1 (t, *J* = 2.2 Hz), 123.0 (t, *J* = 5.8 Hz), 117.7 (t, *J* = 244.3 Hz), 113.6 (t, *J* = 4.9 Hz), 100.9; HRMS (ESI) *m*/*z* [M+H]^+^ calcd for C_13_H_7_Br_2_F_2_N_2_OS: 434.8608; found: 434.8598.

4-(Trifluoromethyl)-2*H*-benzo[4,5]thiazolo[3,2-*a*]pyrimidin-2-one (**3h**).

Compound **3h** was obtained as a yellow solid with a yield of 61%. m.p. 206–208 °C; ^1^H NMR (300 MHz, CDCl_3_): δ 7.90 (dd, *J* = 8.2, 1.8 Hz, 1H), 7.73 (dd, *J* = 7.5, 1.8 Hz, 1H), 7.54 (ddd, *J* = 8.2, 7.5, 1.8 Hz, 1H), 7.53 (ddd, *J* = 8.2, 7.5, 1.8 Hz, 1H), 6.94 (s, 1H); ^19^F NMR (282 MHz, CDCl_3_): δ −62.23; ^13^C NMR (75 MHz, CDCl_3_): δ 166.3, 165.0, 136.1 (q, *J* = 37.0 Hz), 134.1, 127.7 (q, *J* = 0.9 Hz), 127.1, 123.7, 123.2, 119.5 (q, *J* = 274.5 Hz), 115.5 (q, *J* = 8.2 Hz), 113.1 (q, *J* = 5.8 Hz); HRMS (ESI) *m*/*z* [M+H]^+^ calcd for C_11_H_6_F_3_N_2_OS: 271.0147; found: 271.0139.

8-Bromo-4-(trifluoromethyl)-2*H*-benzo[4,5]thiazolo[3,2-*a*]pyrimidin-2-one (**3i**).

Compound **3i** was obtained as an orange solid with a yield of 60%. m.p. 232–234 °C; ^1^H NMR (300 MHz, CDCl_3_): δ 7.86 (d, *J* = 1.5 Hz, 1H), 7.75 (d, *J* = 9.5 Hz, 1H), 7.67 (dd, *J* = 9.5, 1.5 Hz, 1H), 6.96 (s,1H); ^19^F NMR (282 MHz, CDCl_3_): δ −62.32; ^13^C NMR (75 MHz, CDCl_3_): δ 165.8, 164.6, 136.0 (q, *J* = 36.7 Hz), 133.1, 130.9, 125.8, 125.6, 120.5, 119.4 (q, *J* = 274.1 Hz), 116.7 (q, *J* = 8.3 Hz), 113.3 (q, *J* = 5.6 Hz); HRMS (ESI) *m*/*z* [M+H]^+^ calcd for C_11_H_5_BrF_3_N_2_OS: 348.9253; found: 348.9247.

5-(Trifluoromethyl)-7*H*-oxazolo[3,2-*a*]pyrimidin-7-one (**3j**).

Compound **3j** was obtained as a white solid with a yield of 68%. m.p. 221–223 °C; ^1^H NMR (300 MHz, CDCl_3_): δ 7.56 (d, *J* = 2.0 Hz, 1H), 7.42–7.38 (m, 1H), 6.78 (s,1H); ^19^F NMR (282 MHz, CDCl_3_): δ −68.13; ^13^C NMR (75 MHz, DMSO-*d_6_*): δ 168.2, 157.7, 136.3, 132.9 (q, *J* = 37.9 Hz), 119.4 (q, *J* = 273.6 Hz), 113.8 (q, *J* = 2.2 Hz), 111.6 (q, *J* = 3.5 Hz); HRMS (ESI) *m*/*z* [M+H]^+^ calcd for C_7_H_4_F_3_N_2_O_2_: 205.0219; found: 205.0215.

5-(Pentafluoroethyl)-7*H*-oxazolo[3,2-*a*]pyrimidin-7-one (**3k**).

Compound **3k** was obtained as a white solid with a yield of 60%. m.p. 170–172 °C; ^1^H NMR (300 MHz, CDCl_3_): δ 7.52 (d, *J* = 2.1 Hz, 1H), 7.41 (br s, 1H), 6.79 (s,1H); ^19^F NMR (282 MHz, CDCl_3_): δ −83.45, −117.31; ^13^C NMR (75 MHz, CDCl_3_): δ 167.3, 157.3, 134.6, 132.4 (t, *J* = 28.0 Hz), 118.0 (qt, *J* = 287.0, 36.2 Hz), 115.0 (t, *J* = 5.0 Hz), 113.0 (t, *J* = 4.4 Hz), 109.7 (tq, *J* = 258.5, 41.1 Hz); HRMS (ESI) *m*/*z* [M+H]^+^ calcd for C_8_H_4_F_5_N_2_O_2_: 255.0187; found: 255.0180.

5-[(4-Bromophenyl)difluoromethyl]-7*H*-oxazolo[3,2-*a*]pyrimidin-7-one (**3l**).

Compound **3l** was obtained as a white solid with a yield of 68%. m.p. 201–203 °C; ^1^H NMR (300 MHz, CDCl_3_): δ 7.72 (d, *J* = 8.5 Hz, 2H), 7.49–7.51 (m, 2H), 7.46 (d, *J* = 8.5 Hz, 2H), 6.13 (s,1H); ^19^F NMR (282 MHz, CDCl_3_): δ −95.59; ^13^C NMR (75 MHz, CDCl_3_): δ 168.6, 157.5, 139.1 (t, *J* = 31.8 Hz), 134.3, 132.7 (3C), 129.8 (t, *J* = 25.6 Hz), 127.7 (t, *J* = 5.7 Hz), 127.0 (t, *J* = 2.2 Hz), 117.4 (t, *J* = 243.4 Hz), 113.5 (t, *J* = 4.7 Hz), 113.3 (t, *J* = 4.2 Hz); HRMS (ESI) *m*/*z* [M+H]^+^ calcd for C_13_H_8_BrF_2_N_2_O_2_: 340.9732; found: 340.9724.

4-(Trifluoromethyl)-2*H*-benzo[4,5]oxazolo[3,2-*a*]pyrimidin-2-one (**3m**).

Compound **3m** was obtained as a white solid with a yield of 61%. m.p. 193–195 °C; ^1^H NMR (300 MHz, DMSO-*d_6_*): δ 7.40 (dt, *J* = 8.2, 1.3 Hz, 1H), 7.34 (td, *J* = 7.7, 1.3 Hz, 1H), 6.97 (dd, *J* = 8.2, 1.3 Hz, 1H), 6.90 (td, *J* = 7.7, 1.3 Hz, 1H), 6.56 (s, 1H); ^19^F NMR (282 MHz, DMSO-*d_6_*): δ −63.90; ^13^C NMR (75 MHz, DMSO-*d_6_*): δ 168.8, 158.3, 154.4, 140.7 (q, *J* = 33.7 Hz), 132.1, 130.6, 122.0, 119.6 (q, *J* = 274.6 Hz), 119.4, 116.8, 109.2 (q, *J* = 4.5 Hz); HRMS (ESI) *m*/*z* [M+H]^+^ calcd for C_11_H_6_F_3_N_2_O_2_: 255.0303; found: 255.0296.

7-Chloro-4-(trifluoromethyl)-2*H*-benzo[4,5]oxazolo[3,2-*a*]pyrimidin-2-one (**3n**).

Compound **3n** was obtained as a white solid with a yield of 56%. m.p. 249–251 °C; ^1^H NMR (300 MHz, MeOH-*d4*): δ 7.44 (q, *J* = 1.2 Hz, 1H), 7.39 (dd, *J* = 8.7, 2.7 Hz, 1H), 6.97 (d, *J* = 8.7 Hz, 1H), 6.62 (s,1H); ^19^F NMR (282 MHz, MeOH-*d4*): δ −66.31; ^13^C NMR (75 MHz, MeOH-*d4*): δ 171.3, 158.7, 153.1, 141.7 (q, *J* = 34.6 Hz), 131.6, 129.5 (q, *J* = 1.0 Hz), 123.3, 122.4, 119.0 (q, *J* = 274.1 Hz), 117.1, 108.0 (q, *J* = 4.6 Hz); HRMS (ESI) *m*/*z* [M+H]^+^ calcd for C_11_H_5_ClF_3_N_2_O_2_: 288.9913; found: 288.9906.

### 3.3. General Procedure for Suzuki-Miyaura Cross-Coupling Reaction: Synthesis of 2-arylated 7H-thiazolo[3,2-a]pyrimidin-7-ones **4a-l**

A sealed tube was charged with fluorinated 2-bromo-7*H*-thiazolo[3,2-*a*]pyrimidin-7-ones **3e-f** 1 mmol (1.0 equiv), boronic acid 1.2 mmol (1.2 equiv), Na_2_CO_3_ 2 mmol (2.0 equiv), and PdCl_2_(PPh_3_)_2_ (10 mol%). The sealed tube and contents were placed under a vacuum and back-filled with argon under a Schlenk line three times. The mixture of 1,4-dioxane/H_2_O (4/1) was then added under argon, and the reaction mixture was heated at 80 °C for 1 h. After the completion of the reaction as monitored by TLC analysis, the solvents were evaporated under reduced pressure, and the crude residue was purified by silica gel column chromatography to give the desired arylated 7H-thiazolo[3,2-a]pyrimidin-7-ones **4a-l**. (^1^H-NMR and ^13^C-NMR of compounds **4a-l** are shown in Appendix A).

2-Phenyl-5-(trifluoromethyl)-7*H*-thiazolo[3,2-*a*]pyrimidin-7-one (**4a**).

The purification of the crude product by chromatography on silica gel is carried out using (PE/EtOAc: 8/2) to afford **4a** as a beige solid with a yield of 91%. m.p. 216–218 °C; ^1^H NMR (300 MHz, CDCl_3_): δ 7.54–7.49 (m, 6H), 6.82 (s, 1H); ^19^F NMR (282 MHz, CDCl_3_): δ −67.81; ^13^C NMR (75 MHz, CDCl_3_): δ 165.7, 165.1, 135.2 (q, *J* = 37.3 Hz), 130.5, 129.6 (2C), 129.2, 128.3, 126.3 (2C), 119.1 (q, *J* = 274.5 Hz), 115.2 (q, *J* = 3.3 Hz), 111.8 (q, *J* = 4.1 Hz); HRMS (ESI) *m*/*z* [M+H]^+^ calcd for C_13_H_8_F_3_N_2_OS: 297.0304; found: 297.0300.

2-(2-Methoxyphenyl)-5-(trifluoromethyl)-7*H*-thiazolo[3,2-*a*]pyrimidin-7-one (**4b**).

The purification of the crude product by chromatography on silica gel is carried out using (PE/EtOAc: 7/3) to afford **4b** as a white solid with a yield of 82%. m.p. 227–229 °C; ^1^H NMR (300 MHz, CDCl_3_): δ 7.85 (q, *J* = 1.5 Hz, 1H), 7.46 (d, *J* = 7.6 Hz, 2H), 7.45 (td, *J* = 7.3, 1.5 Hz, 2H) 7.12–7.05 (m, 2H), 6.81 (s, 1H), 4.00 (s, 3H); ^19^F NMR (282 MHz, CDCl_3_): δ −67.91; ^13^C NMR (75 MHz, CDCl_3_): δ 165.7, 165.4, 156.4, 135.0 (q, *J* = 37.1 Hz), 131.3, 128.1, 125.1, 121.5, 119.2 (q, *J* = 274.4 Hz), 117.8 (q, *J* = 3.5 Hz), 117.2, 111.8 (q, *J* = 4.1 Hz), 111.7, 55.8; HRMS (ESI) *m*/*z* [M+H]^+^ calcd for C_14_H_10_F_3_N_2_O_2_S: 327.0410; found: 327.0404.

2-(3-Methoxyphenyl)-5-(trifluoromethyl)-7*H*-thiazolo[3,2-*a*]pyrimidin-7-one (**4c**).

The purification of the crude product by chromatography on silica gel is carried out using (PE/EtOAc: 7/3) to afford **4c** as a white solid with a yield of 92%. m.p. 193–195 °C; ^1^H NMR (300 MHz, CDCl_3_): δ 7.49 (q, *J* = 1.5 Hz, 1H), 7.43 (td, *J* = 7.7, 0.9 Hz, 1H), 7.11 (dt, *J* = 7.7, 0.9 Hz, 1H), 7.04–7.01 (m, 2H), 6.82 (s, 1H), 3.90 (s, 3H); ^19^F NMR (282 MHz, CDCl_3_): δ = −67.80; ^13^C NMR (75 MHz, CDCl_3_): δ 165.6, 165.1, 160.4, 135.2 (q, *J* = 37.2 Hz), 130.8, 129.5, 129.0, 119.1 (q, *J* = 274.5 Hz), 118.7, 115.5, 115.4 (q, *J* = 3.5 Hz), 112.4, 111.9 (q, *J* = 3.9 Hz), 55.6; HRMS (ESI) *m*/*z* [M+H]^+^ calcd for C_14_H_10_F_3_N_2_O_2_S: 327.0410; found: 327.0405. 

2-(4-Methoxyphenyl)-5-(trifluoromethyl)-7*H*-thiazolo[3,2-*a*]pyrimidin-7-one (**4d**).

The purification of the crude product by chromatography on silica gel is carried out using (PE/EtOAc: 7/3) to afford **4d** as a beige solid with a yield of 95%. m.p. 181–183 °C; ^1^H NMR (300 MHz, CDCl_3_): δ 7.45 (d, *J* = 9 Hz, 2H), 7.39 (q, *J* = 1.5 Hz, 1H), 7.02 (d, *J* = 9 Hz, 2H), 6.81 (s, 1H), 3.89 (s, 3H); ^19^F NMR (282 MHz, CDCl_3_): δ −67.87; ^13^C NMR (75 MHz, CDCl_3_): δ 165.5, 165.1, 161.3, 135.1 (q, *J* = 37.0 Hz), 128.5, 127.7 (2C), 120.6, 119.2 (q, *J* = 274.7 Hz), 115.0 (2C), 114.0 (q, *J* = 3.2 Hz), 111.7 (q, *J* = 4.0 Hz), 55.5; HRMS (ESI) *m*/*z* [M+H]^+^ calcd for C_14_H_10_F_3_N_2_O_2_S: 327.0410; found: 327.0405.

2-(4-Methoxyphenyl)-5-(pentafluoroethyl)-7*H*-thiazolo[3,2-*a*]pyrimidin-7-one (**4e**).

The purification of the crude product by chromatography on silica gel is carried out using (PE/EtOAc: 8/2) to afford **4e** as a beige solid with a yield of 55%. m.p. 201–203 °C; ^1^H NMR (300 MHz, CDCl_3_): δ 7.46 (s, 1H), 7.43 (d, *J* = 8.8 Hz, 2H), 7.01 (d, *J* = 8.8 Hz, 2H), 6.79 (s, 1H); ^19^F NMR (282 MHz, CDCl_3_): δ −82.28, −115.50; ^13^C NMR (75 MHz, CDCl_3_): δ 166.1, 164.8, 161.3, 134.5 (t, *J* = 26.9 Hz), 128.6, 127.7 (2C), 120.6, 118.1 (qt, *J* = 287.5, 36.1 Hz), 115.0 (2C), 114.9 (td, *J* = 7.1, 1.7 Hz), 114.6 (t, *J* = 5.9 Hz), 110.1 (tq, *J* = 258.6 Hz, *J* = 40.9 Hz), 55.5; HRMS (ESI) *m*/*z* [M+H]^+^ calcd for C_15_H_10_F_5_N_2_O_2_S: 377.0378; found: 377.0372.

2-(2,4-Dimethoxyphenyl)-5-(trifluoromethyl)-7*H*-thiazolo[3,2-*a*]pyrimidin-7-one (**4f**).

The purification of the crude product by chromatography on silica gel is carried out using (PE/EtOAc: 7/3) to afford **4f** as a solid beige with a yield of 98%. m.p. 260–262 °C; ^1^H NMR (300 MHz, CDCl_3_): δ 7.73 (q, *J* = 1.5 Hz, 1H), 7.37 (d, *J* = 8.5 Hz, 1H), 6.79 (s, 1H), 6.64–6.57 (m, 2H), 3.97 (s, 3H), 3.89 (s, 3H); ^19^F NMR (282 MHz, CDCl_3_): δ −67.97; ^13^C NMR (75 MHz, CDCl_3_): δ 165.6, 165.5, 162.4, 157.7, 135.5 (q, *J* = 37.2 Hz), 132.2, 129.1, 125.4, 119.2 (q, *J* = 274.4 Hz), 116.2 (q, *J* = 3.5 Hz), 111.6 (q, *J* = 3.8 Hz), 110.1, 99.1, 55.8, 55.6; HRMS (ESI) *m*/*z* [M+H]^+^ calcd for C_15_H_12_F_3_N_2_O_3_S: 357.0515; found: 357.0510.

2-(2,3-Dihydrobenzo[*b*][1,4]dioxin-6-yl)-5-(trifluoromethyl)-7*H*-thiazolo[3,2-*a*]pyrimidin-7-one (**4g**).

The purification of the crude product by chromatography on silica gel is carried out using (PE/EtOAc: 7/3) to afford **4g** as a solid white with a yield of 60%. m.p. 258–260 °C; ^1^H NMR (300 MHz, MeOH-*d_4_*): δ 7.91 (q, *J* = 1.5 Hz, 1H), 7.23 (d, *J* = 2.3 Hz, 1H), 7.18 (dd, *J* = 8.4, 2.3 Hz, 1H), 6.97 (d, *J* = 8.4 Hz, 1H), 6.90 (s, 1H), 4.32 (s, 4H); ^19^F NMR (282 MHz, MeOH-*d_4_*): δ −69.44; ^13^C NMR (75 MHz, CDCl_3_): δ 165.5, 165.2, 145.6, 144.3, 135.1 (q, *J* = 37.3 Hz), 128.9, 121.4, 119.6, 119.1 (q, *J* = 274.6 Hz), 118.5, 115.2, 114.3 (q, *J* = 3.3 Hz), 111.7 (q, *J* = 4.0 Hz), 64.5, 64.4; HRMS (ESI) *m*/*z* [M+H]^+^ calcd for C_15_H_10_F_3_N_2_O_3_S: 355.0359; found: 355.0354.

2-(3-Aminophenyl)-5-(trifluoromethyl)-7*H*-thiazolo[3,2-*a*]pyrimidin-7-one (**4h**).

The purification of the crude product by chromatography on silica gel is carried out using (PE/EtOAc: 5/5) to afford **4h** as a beige solid with a yield of 52%. m.p. 253–255 °C; ^1^H NMR (300 MHz, DMSO-*d_6_*): δ 7.93 (s, 1H), 7.14 (t, *J* = 8.0 Hz, 1H), 6.92 (d, *J* = 8.0 Hz, 1H), 6.86 (s, 1H), 6.84 (q, *J* = 1.6 Hz, 1H), 6.65 (dd, *J* = 8.0, 1.6 Hz, 1H), 5.39 (s, 2H); ^19^F NMR (282 MHz, DMSO-*d_6_*): δ −66.88; ^13^C NMR (75 MHz, DMSO-*d_6_*): δ 165.7, 165.2, 150.0, 135.1 (q, *J* = 36.9 Hz), 130.4, 129.2, 128.1, 119.5 (q, *J* = 274.6 Hz), 116.9 (q, *J* = 2.4 Hz), 115.8, 114.0, 111.8 (q, *J* = 3.8 Hz), 111.4; HRMS (ESI) *m*/*z* [M+H]^+^ calcd for C_13_H_9_F_3_N_3_OS: 312.0340; found: 312.0336.

5-(Trifluoromethyl)-2-(3-(trifluoromethyl)phenyl)-7*H*-thiazolo[3,2-*a*]pyrimidin-7-one (**4i**).

The purification of the crude product by chromatography on silica gel is carried out using (PE/EtOAc: 6/4) to afford **4i** as a white solid with a yield of 75%. m.p. 241–243 °C; ^1^H NMR (300 MHz, MeOH-*d_4_*): δ = 8.26 (q, *J* = 1.5 Hz, 1H), 8.09 (s, 1H), 7.99 (d, *J* = 7.7 Hz, 1H), 7.82 (d, *J* = 7.7 Hz, 1H), 7.74 (t, *J* = 7.7 Hz, 1H), 6.81 (s, 1H); ^19^F NMR (282 MHz, MeOH-*d_4_*): δ = −64.29, −69.33; ^13^C NMR (75 MHz, DMSO-*d_6_*): δ = 165.9, 165.2, 135.3 (q, *J* = 37.0 Hz), 131.0, 130.9, 130.6 (q, *J* = 32.2 Hz), 130.2, 126.5 (q, *J* = 3.8 Hz), 125.6, 124.3 (q, *J* = 272.8 Hz), 123.4 (q, *J* = 3.7 Hz), 119.8 (q, *J* = 2.7 Hz), 119.4 (q, *J* = 274.5 Hz), 112.1 (q, *J* = 4.1 Hz); HRMS (ESI) *m*/*z* [M+H]^+^ calcd for C_14_H_7_F_6_N_2_OS: 365.0178; found: 365.0174.

5-(Trifluoromethyl)-2-(4-(trifluoromethyl)phenyl)-7*H*-thiazolo[3,2-*a*]pyrimidin-7-one (**4j**).

The purification of the crude product by chromatography on silica gel is carried out using (PE/EtOAc: 6/4) to afford **6e** as a white solid with a yield of 57%. m.p. 262–264 °C; ^1^H NMR (300 MHz, MeOH-*d_4_*): δ 8.22 (q, *J* = 1.5 Hz, 1H), 7.96 (d, *J* = 8.2 Hz, 2H), 7.84 (d, *J* = 8.2 Hz, 2H), 6.95 (s, 1H); ^19^F NMR (282 MHz, MeOH-*d_4_*): δ = −64.46, −69.32; ^13^C NMR (75 MHz, DMSO-*d_6_*): δ = 166.0, 165.2, 135.3 (q, *J* = 37.0 Hz), 133.1 (q, *J* = 1.4 Hz), 130.0 (q, *J* = 32.3 Hz), 127.6 (2C), 126.6 (q, *J* = 3.8 Hz, 2C), 125.5, 124.4 (q, *J* = 272.1 Hz), 120.1 (q, *J* = 3.0 Hz), 119.4 (q, *J* = 274.6 Hz), 112.1 (q, *J* = 3.9 Hz); HRMS (ESI) *m*/*z* [M+H]^+^ calcd for C_14_H_7_F_6_N_2_OS: 365.0178; found: 365.0172.

Ethyl 4-(7-oxo-5-(trifluoromethyl)-7*H*-thiazolo[3,2-*a*]pyrimidin-2-yl)benzoate (**4k**).

The purification of the crude product by chromatography on silica gel is carried out using (PE/EtOAc: 7/3) to afford **4k** as a white solid with a yield of 60%. m.p. 254–256 °C; ^1^H NMR (300 MHz, CDCl_3_): δ 8.18 (d, *J* = 8.5 Hz, 2H), 7.60 (q, *J* = 1.5 Hz, 1H), 7.59 (d, *J* = 8.5 Hz, 2H), 6.84 (s, 1H), 4.44 (q, *J* = 7.2 Hz, 2H), 1.44 (t, *J* = 7.2 Hz, 3H); ^19^F NMR (282 MHz, CDCl_3_): δ −67.70; ^13^C NMR (75 MHz, DMSO-*d_6_*): δ 165.9, 165.5, 165.2, 135,2 (q, *J* = 37.1 Hz), 133.9, 130.9, 130.4 (2C), 127 (2C), 125.9, 119.9 (q, *J* = 3.2 Hz), 119.4 (q, *J* = 274.7 Hz), 112.2 (q, *J* = 4.0 Hz), 61.5, 14.6; HRMS (ESI) *m*/*z* [M+H]^+^ calcd for C_16_H_11_F_3_N_2_O_3_S: 369.0442; found: 369.0437.

2-(Thiophen-3-yl)-5-(trifluoromethyl)-7*H*-thiazolo[3,2-*a*]pyrimidin-7-one (**4l**).

The purification of the crude product by chromatography on silica gel is carried out using (PE/EtOAc: 5/5) to afford **4l** as a white solid with a yield of 95%. m.p. 222–224 °C; ^1^H NMR (300 MHz, MeOH-*d_4_*): δ 8.00 (q, *J* = 1.5 Hz, 1H), 7.90 (dd, *J* = 2.9, 1.3 Hz, 1H), 7.63 (dd, *J* = 5.2, 2.9 Hz, 1H), 7.55 (dd, *J* = 5.2, 1.3 Hz, 1H), 6.91 (s, 1H); ^19^F NMR (282 MHz, MeOH-*d_4_*): δ −69.44; ^13^C NMR (75 MHz, MeOH-*d_4_*): δ 167.1, 166.4, 136.2 (q, *J* = 37.4 Hz), 129.1, 127.8, 125.1, 124.6, 124.4, 119.1 (q, *J* = 273.9 Hz), 116.7 (q, *J* = 3.2 Hz), 110.9 (q, *J* = 4.1 Hz); HRMS (ESI) *m*/*z* [M+H]^+^ calcd for C_11_H_6_F_3_N_2_OS_2_: 302.9868; found: 302.9863.

### 3.4. General Procedure for Sonogashira Cross-Coupling Reaction: Synthesis of 2-alkynylated 5-(trifluoromethyl)-7H-thiazolo[3,2-a]pyrimidin-7-ones **5a-h**

Then, 2-Bromo-5-(trifluoromethyl)-7*H*-thiazolo[3,2-*a*]pyrimidin-7-one **3e** 1 mmol (1 equiv), alkyne 1.5 mmol (1.5 equiv), CuI (10 mol%), Et_3_N 2 mmol (2 equiv), and PdCl_2_(PPh_3_)_2_ (5 mol%) were added in an oven-dried 10 mL sealed tube. The reaction tube was placed under a vacuum and backfilled with argon three times. Then anhydrous DMF (4.0 mL) was added to the tube via a microsyringe, and the reaction mixture was refluxed at 80 °C for 2 h. Upon the consumption of the starting material, **3e** (determined by TLC), the solvent was evaporated under reduced pressure, and the crude residue was purified by silica gel column chromatography to give the desired alkynylated 5-(trifluoromethyl)-7*H*-thiazolo[3,2-*a*]pyrimidin-7-ones **5a-h**. (^1^H-NMR and ^13^C-NMR of compounds **5a-h** are shown in Appendix A.)

2-(Phenylethynyl)-5-(trifluoromethyl)-7*H*-thiazolo[3,2-*a*]pyrimidin-7-one (**5a**).

The purification of the crude product by chromatography on silica gel is carried out using (PE/EtOAc: 8/2) to afford **5a** as a beige solid with a yield of 68%. m.p. 181–183 °C; ^1^H NMR (300 MHz, CDCl_3_): δ 7.56 (q, *J* = 1.4 Hz, 1H), 7.57–7.51 (m, 3H), 7.47–7.39 (m, 3H), 6.81 (s, 1H); ^19^F NMR (282 MHz, CDCl_3_): δ −67.91; ^13^C NMR (75 MHz, CDCl_3_): δ 165.1, 164.9, 134.9 (q, *J* = 37.8 Hz), 131.8 (2C), 130.1, 128.7 (2C), 123.3 (q, *J* = 3.3 Hz), 120.7, 119.0 (q, *J* = 274.5 Hz), 111.9 (q, *J* = 3.6 Hz), 110.1, 99.7, 76.0; HRMS (ESI) *m*/*z* [M+H]^+^ calcd for C_15_H_8_F_3_N_2_OS: 321.0304; found: 321.0300.

2-[(4-Methoxyphenyl)ethynyl]-5-(trifluoromethyl)-7*H*-thiazolo[3,2-*a*]pyrimidin-7-one (**5b**).

The purification of the crude product by chromatography on silica gel is carried out using (PE/EtOAc: 8/2) to afford **5b** as a beige solid with a yield of 65%. m.p. 213–215 °C; ^1^H NMR (300 MHz, CDCl_3_): δ 7.49 (d, *J* = 8.9 Hz, 2H), 7.46 (q, *J* = 1.5 Hz, 1H), 6.94 (d, *J* = 8.9 Hz, 2H), 6.80 (s, 1H), 3.87 (s, 1H); ^19^F NMR (282 MHz, CDCl_3_): δ = −67.95; ^13^C NMR (75 MHz, CDCl_3_): δ 165.1, 164.9, 161.1, 134.8 (q, *J* = 37.4 Hz), 133.5 (2C), 122.7 (q, *J* = 3.4 Hz), 119.0 (q, *J* = 274.4 Hz), 114.4 (2C), 112.6, 111.8 (q, *J* = 3.4 Hz), 110.5, 100.1, 75.0, 55.4; HRMS (ESI) *m*/*z* [M+H]^+^ calcd for C_16_H_10_F_3_N_2_O_2_S: 351.0410; found: 351.0406.

2-[(4-Chlorophenyl)ethynyl]-5-(trifluoromethyl)-7*H*-thiazolo[3,2-*a*]pyrimidin-7-one (**5c**).

The purification of the crude product by chromatography on silica gel is carried out using (PE/EtOAc: 8/2) to afford **5c** as a yellow solid with a yield of 58%. m.p. 243–245 °C; ^1^H NMR (300 MHz, CDCl_3_): δ 7.51 (q, *J* = 1.5 Hz, 1H), 7.48 (d, *J* = 8.7 Hz, 2H), 7.41 (d, *J* = 8.7 Hz, 2H), 6.81 (s, 1H); ^19^F NMR (282 MHz, CDCl_3_): δ = −67.90; ^13^C NMR (75 MHz, CDCl_3_): δ 165.1, 164.8, 136.5, 134.8 (q, *J* = 37.5 Hz), 133.0 (2C), 129.2 (2C), 123.5 (q, *J* = 3.3 Hz), 119.1, 119.0 (q, *J* = 274.7 Hz), 111.9 (q, *J* = 3.7 Hz), 109.7, 98.4, 76.9; HRMS (ESI) *m*/*z* [M+H]^+^ calcd for C_15_H_7_ClF_3_N_2_OS: 354.9914; found: 354.9910.

2-[(4-Fluorophenyl)ethynyl]-5-(trifluoromethyl)-7*H*-thiazolo[3,2-*a*]pyrimidin-7-one (**5d**).

The purification of the crude product by chromatography on silica gel is carried out using (PE/EtOAc: 8/2) to afford **5d** as a yellow solid with a yield of 85%. m.p. 203–205 °C; ^1^H NMR (300 MHz, CDCl_3_): δ 7.55 (dd, *J* = 8.7, 5.3 Hz, 2H), 7.50 (q, *J* = 1.5 Hz, 1H), 7.12 (t, *J* = 8.7 Hz, 2H), 6.81 (s, 1H); ^19^F NMR (282 MHz, CDCl_3_): δ −67.91, -107.19; ^13^C NMR (75 MHz, CDCl_3_): δ 165.1, 164.8, 163.6 (d, *J* = 253.0 Hz), 134.9 (q, *J* = 37.5 Hz), 134.0, 133.9, 123.4 (q, *J* = 3.2 Hz), 119.0 (q, *J* = 274.7 Hz), 116.8 (d, *J* = 3.6 Hz), 116.4, 116.1, 111.9 (q, *J* = 3.9 Hz), 109.9, 98.5, 75.8 (d, *J* = 1.4 Hz); HRMS (ESI) *m*/*z* [M+H]^+^ calcd for C_15_H_7_F_4_N_2_OS: 339.0210; found: 339.0207.

2-[(2-Fluorophenyl)ethynyl]-5-(trifluoromethyl)-7*H*-thiazolo[3,2-*a*]pyrimidin-7-one (**5e**).

The purification of the crude product by chromatography on silica gel is carried out using (PE/EtOAc: 8/2) to afford **5e** as a yellow solid with a yield of 68%. m.p. 176–178 °C; ^1^H NMR (300 MHz, CDCl_3_): δ 7.56–7.51 (m, 2H), 7.50–7.42 (m, 1H), 7.21 (t, *J* = 7.7 Hz, 1H), 7.16 (d, *J* = 9.2 Hz, 1H), 6.82 (s, 1H); ^19^F NMR (282 MHz, CDCl_3_): δ −67.88, −108.23; ^13^C NMR (75 MHz, CDCl_3_): δ 165.1, 164.8, 162.8 (d, *J* = 254.2 Hz), 134.4 (q, *J* = 37.6 Hz), 133.4, 132.0 (d, *J* = 8.2 Hz), 124.3 (d, *J* = 3.8 Hz), 123.8 (q, *J* = 3.3 Hz), 119.0 (q, *J* = 274.7 Hz), 116.0 (d, *J* = 20.4 Hz), 111.9 (q, *J* = 3.9 Hz), 109.6 (d, *J* = 2.5 Hz), 109.4, 93.0, 80.7 (d, *J* = 3.3 Hz); HRMS (ESI) *m*/*z* [M+H]^+^ calcd for C_15_H_7_F_4_N_2_OS: 339.0210; found: 339.0206.

2-(Pyridin-2-ylethynyl)-5-(trifluoromethyl)-7*H*-thiazolo[3,2-*a*]pyrimidin-7-one (**5f**).

The purification of the crude product by chromatography on silica gel is carried out using (PE/EtOAc: 6/4) to afford **5f** as a brown solid with a yield of 92%. m.p. 193–195 °C; ^1^H NMR (300 MHz, CDCl_3_): δ 8.70 (s, 1H), 7.80 (td, *J* = 7.8, 1.5 Hz, 1H), 7.60 (q, *J* = 1.5 Hz, 1H), 7.60 (d, *J* = 7.8 Hz, 1H), 7.40–7.38 (m, 1H), 6.82 (s, 1H); ^19^F NMR (282 MHz, CDCl_3_): δ −67.92; ^13^C NMR (75 MHz, CDCl_3_): δ 165.2, 164.8, 150.5, 141.2, 136.6, 134.9 (q, *J* = 37.9 Hz), 127.5, 124.9 (q, *J* = 3.4 Hz), 124.3 (q, *J* = 3.4 Hz), 118.9 (q, *J* = 274.7 Hz), 112.0 (q, *J* = 3.9 Hz), 109.0, 97.9, 75.5; HRMS (ESI) *m*/*z* [M+H]^+^ calcd for C_14_H_7_F_3_N_3_OS: 322.0256; found: 322.0252.

2-(Cyclohexylethynyl)-5-(trifluoromethyl)-7*H*-thiazolo[3,2-*a*]pyrimidin-7-one (**5g**).

The purification of the crude product by chromatography on silica gel is carried out using (PE/EtOAc: 9/1) to afford **5g** as a beige solid with a yield of 75%. m.p. 146–148 °C; ^1^H NMR (300 MHz, CDCl_3_): δ 7.34 (q, *J* = 1.5 Hz, 1H), 6.77 (s, 1H), 2.68–2.60 (m, 1H), 1.90–1.87 (m, 2H), 1.79–1.71 (m, 2H), 1.62–1.48 (m, 3H), 1.44–1.34 (m, 3H); ^19^F NMR (282 MHz, CDCl_3_): δ −67.97; ^13^C NMR (75 MHz, CDCl_3_): δ 165.1, 134.4 (q, *J* = 37.4 Hz), 128.5, 122.6 (q, *J* = 3.3 Hz), 119.0 (q, *J* = 274.5 Hz), 111.7 (q, *J* = 4.0 Hz), 110.9, 106.3, 67.7, 31.9 (2C), 29.9, 25.6, 24.7 (2C); HRMS (ESI) *m*/*z* [M+H]^+^ calcd for C_15_H_14_F_3_N_2_OS: 327.0773, found: 327.0769.

2-(Hex-1-yn-1-yl)-5-(trifluoromethyl)-7*H*-thiazolo[3,2-*a*]pyrimidin-7-one (**5h**).

The purification of the crude product by chromatography on silica gel is carried out using (PE/EtOAc: 9/1) to afford **5h** as a yellow solid with a yield of 55%. m.p. 145–147 °C; ^1^H NMR (300 MHz, CDCl_3_): δ 7.34 (q, *J* = 1.5 Hz, 1H), 6.77 (s, 1H), 2.47 (t, *J* = 7.2 Hz, 2H), 1.66–157 (m, 2H), 1.53–1.41 (m, 2H), 0.97 (t, *J* = 7.2 Hz, 3H); ^19^F NMR (282 MHz, CDCl_3_): δ −67.98; ^13^C NMR (75 MHz, CDCl_3_): δ 165.1, 135.3 (q, *J* = 37.3 Hz), 128.5, 122.7 (q, *J* = 3.4 Hz), 119.0 (q, *J* = 274.6 Hz), 111.7 (q, *J* = 3.8 Hz), 110.8, 102.6, 67.8, 30.0, 22.0, 19.4, 13.5; HRMS (ESI) *m*/*z* [M+H]^+^ calcd for C_13_H_11_F_3_N_2_OS: 301.0617; found: 301.0614.

## 4. Conclusions

In conclusion, we have developed a simple and convenient method for straightforward access to an important range of fluorinated thiazolo[3,2-*a*]pyrimidin-7-ones and oxazolo[3,2-*a*]pyrimidin-7-ones. We first established a concise one-pot strategy for the synthesis of 5-fluoroalkylated thiazolo -and oxazolo[3,2-*a*]pyrimidin-7-ones by condensation of 2-amino thiazole or 2-amino oxazole derivatives with fluorinated ethyl propiolates. The synthesized 2-bromo-5-trifluoromethyl thiazolo[3,2-*a*]pyrimidin-7-ones were used as building blocks for the synthesis of a series of new 2-arylated and 2-alkynylated thiazolo[3,2-*a*]pyrimidin-7-ones containing a fluoroalkyl group. A preliminary biological evaluation carried out on a few synthesized compounds on human hMAO-A and hMAO-B showed that compound **3g** exhibits a selective micromolar inhibition of hMAO-A, which is a promising target in the symptomatic treatment and potentially disease-modifying treatment of neurodegenerative disorders.

Further exploration of this strategy and further evaluation of the biological potential of the synthesized compounds are currently under investigation in our laboratory.

## Data Availability

Not available.

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
