# Peer review of "Simple and Expedient Access to Novel Fluorinated Thiazolo- and Oxazolo[3,2-a]pyrimidin-7-one Derivatives and Their Functionalization via Palladium-Catalyzed Reactions"

_molecules, 2022, doi:10.3390/molecules27093013_

Round 1
Reviewer 1 Report
This manuscript -by Jismy, Abarbri and coworkers- presents a new one-pot synthesis for novel fluorinated thiazolo- and oxazolo[3,2-a]pyrimidin-7-ones. These fluorinated cyclized compounds were formed in good yields without any catalyst. In palladium-catalyzed Suzuki and Sonogashira reactions formed several new well-characterized molecular libraries.
The manuscript is well-written and deserves publication once the following points are considered:
- Why were these salts used as catalysts (Ag(I), Cu(I), Zn(II), Pd(II) etc.) in Table 1? Why these are useful in organic chemistry?
- According to "Instructions for Authors" summary of the diffraction parameters for a 3a should be included in the main body of the article. In this version of the paper all parameters, diffractometer data and software are missing and should be inserted into the paper. CCDC number is required, too.
- Why only the 3a was confirmed with solid-state structure?
Overall, this is a good synthetic work leading to a number of fluorinated compounds.
Author Response
Dear Reviewer,
Thank you for your decision and your comments .
Please find below the answers following all the remarks and comments from the referees in response to the revised manuscript “Simple and expedient access to novel fluorinated thiazolo- and oxazolo[3,2-a]pyrimidin-7-one derivatives and their functionalization via palladium-catalyzed reactions”
All corrections and changes are marked in red color in the revised manuscript. We have also added some arguments in connection with the general remarks of the referees.
Reviewer 1:
- Comments and Suggestions for Authors:
Why were these salts used as catalysts (Ag(I), Cu(I), Zn(II), Pd(II) etc.) in Table 1? Why these are useful in organic chemistry?
Response:
In the case of a reactivity involving a nucleophilic attack of a heteroatom on a triple bond subsequently inducing intramolecular cyclization, it is possible to use these metals (Ag(I), Pd(II), Cu(I), Zn(II), Pd(II) etc.) as a catalyst to enhance the reactivity of these alkynes and therefore promote the reaction process. The formation of a pi complex (between the metal and the triple bond) explains this. In the case of activated alkynes (which is our case here), the increase in the reactivity of the triple bond is the result of a coordination of the metal with the heteroatom, subsequently facilitating the regioselective nucleophilic attack of nitrogen on the triple bond. This phenomenon has been widely described in the literature, for example in the following references:
Adv. Synth. Catal. 2022, 364, 466–4 ; Org. Biomol. Chem. 2022, 20, 1518–1531 ; Asian J. Org. Chem. 2018, 7, 123-127 ; J. Org. Chem. 2021, 86, 16940-16947 ; Molecules 2021, 26, 2318 ; Eur. J. Org. Chem. 2015, 3251–3265 ; Heterocycles 2007, 73, 187–190 .
- Comments and Suggestions for Authors:
According to "Instructions for Authors" summary of the diffraction parameters for a 3a should be included in the main body of the article. In this version of the paper all parameters, diffractometer data and software are missing and should be inserted into the paper. CCDC number is required, too.
Response:
The reviewer is right. This is an oversight. All requested diffraction parameters requested for a 3a are listed in Table 2 in the supporting information part. The CCDC has been added in reference 49 of the revised manuscript.
- Comments and Suggestions for Authors:
Why only the 3a was confirmed with solid-state structure?
Response:
We understand the comment of the reviewer. Insofar as it is about a similar approach for all the cases of cyclocondensation described in this article, it seemed to us sufficient to be based on the structure of the compound 3a, used like model in the study of optimization of reaction conditions, to generalize our approach. Also, the spectroscopic analysis (1H NMR, 13C NMR, 19F NMR and HRMS) are in perfect agreement with the proposed structures.
Reviewer 2 Report
The submitted manuscript is well written. The authors synthesized a number of new compounds. All the compounds were characterized by 1H, 13C, and 19F NMR spectra, melting points, and HRMS. As a bonus one single crystal ORTEP diagram is presented. The potential biological activity was tested for the selected compound and the others are being tested.
The presented manuscript is publishable in the Molecules journal after minor revision.
- In the experimental part, it would be appropriate to follow the order e.g., yellow solid instead of solid yellow. Please unify.
- In the experimental part, it would be useful to indicate the amount of substance used, e.g., 1 mmol (1.0 eq) instead of only (1.0 equiv). Please fill in.
- Why was the elemental microanalysis not performed? It should be a standard, especially for the new compounds. According to NMR, some unspecified impurities are present. Elemental analysis, unlike HRMS or NMR, shows purity. Please perform an elemental analysis.
- The 19F NMR spectra are, in my opinion, incorrectly phased. It looks confusing when some signals go below the baseline (like APT in 13C). If it is the aim, please specify the experiment in more detail or correct the phasing. I understand some phasing is impracticable, then it should be commented on.
- The ORTEP diagram is the only crystallographic information in the manuscript. Due to the fact that this is a new substance not yet described. The crystallographic information should be mentioned at a minimum in the supplementary part. Standard crystallographic information e.g., unit cell, system, space group, volume, Z value, density, the temperature of measurement, numbers of observed and unique reflections, and mainly the CCDC deposition number must be specified. I was not able to find the CCDC (obtain the deposited cif file). Due to the fact that the reviewed manuscript should be checked also from the crystallographic point of view before possible publication. I must check the deposited CCDC information which I was not able to obtain. Please provide the CCDC number correctly, before the second revision.
After these minimal corrections, the article will be publishable in Molecules.
Author Response
Dear Reviewer,
Thank you for your decision and your comments .
Please find below the answers following all the remarks and comments from the referees in response to the revised manuscript “Simple and expedient access to novel fluorinated thiazolo- and oxazolo[3,2-a]pyrimidin-7-one derivatives and their functionalization via palladium-catalyzed reactions”
All corrections and changes are marked in red color in the revised manuscript. We have also added some arguments in connection with the general remarks of the referees.
- Comments and Suggestions for Authors:
In the experimental part, it would be appropriate to follow the order e.g., yellow solid instead of solid yellow. Please unify.
Response:
Thank you for this remark. We have corrected these errors
- Comments and Suggestions for Authors:
In the experimental part, it would be useful to indicate the amount of substance used, e.g., 1 mmol (1.0 eq) instead of only (1.0 equiv). Please fill in
Response:
Thank you to the Reviewer for this remark. We have added the requested suggestions.
- Comments and Suggestions for Authors:
Why was the elemental microanalysis not performed? It should be a standard, especially for the new compounds. According to NMR, some unspecified impurities are present. Elemental analysis, unlike HRMS or NMR, shows purity. Please perform an elemental analysis.
Response:
We understand the referee's comment. He is absolutely right. In our case, we assumed that the results of the 1H NMR, 19F NMR, 13C NMR and HRMS analyzes are sufficiently reliable to attest to the sufficient purity of our products. Unfortunately, we do not have a microanalysis center. We have to do them elsewhere and it usually takes a long time to get the results.
- Comments and Suggestions for Authors:
The 19F NMR spectra are, in my opinion, incorrectly phased. It looks confusing when some signals go below the baseline (like APT in 13C). If it is the aim, please specify the experiment in more detail or correct the phasing. I understand some phasing is impracticable, then it should be commented on.
Response:
The reviewer is right. The phasing of the 19F NMR spectra have been corrected and adjusted in the supporting information part.
- Comments and Suggestions for Authors:
The ORTEP diagram is the only crystallographic information in the manuscript. Due to the fact that this is a new substance not yet described. The crystallographic information should be mentioned at a minimum in the supplementary part. Standard crystallographic information e.g., unit cell, system, space group, volume, Z value, density, the temperature of measurement, numbers of observed and unique reflections, and mainly the CCDC deposition number must be specified. I was not able to find the CCDC (obtain the deposited cif file). Due to the fact that the reviewed manuscript should be checked also from the crystallographic point of view before possible publication. I must check the deposited CCDC information which I was not able to obtain. Please provide the CCDC number correctly, before the second revision.
Response:
The reviewer is right. This is an oversight. All requested diffraction parameters for a 3a are listed in Table 2 in the supporting information part. The CCDC has been added in reference 49 of the revised manuscript.